# Governance and Practices for Achieving Sustainable and Resilient Urban Water Services

Jyrki Laitinen [1,*] , Tapio S. Katko [2] , Jarmo J. Hukka [2], Petri Juuti [2] and Riikka Juuti [2]

1   Finnish Environment Institute SYKE, FI-00790 Helsinki, Finland
2   Faculty of Built Environment, Tampere University, 33100 Tampere, Finland; tapio.katko@tuni.fi (T.S.K.);
    omraj@jarmohukka.fi (J.J.H.); petri.juuti@tuni.fi (P.J.); riikka.juuti@tuni.fi (R.J.)
*   Correspondence: jyrki.laitinen@syke.fi; Tel.: +358-295-251-346

**Abstract:** Urban water services can be considered a part of municipal services, including the technical solution from water source to water treatment and distribution, and also wastewater collection, treatment, and discharge back to natural waters. The main aspect is how comprehensive water services concerning the whole society should be considered in urban development. This article emphasizes the necessary role of water services in community technical services and analyzes its critical functions. To keep urban water services on a sound environmental and health level even in externally or internally changing situations is considered sustainable and resilient. In our study, we used sequential PESTEL SWOT analysis to review the results of previous studies. The conclusions and recommendations are based on practices and governance of urban water services in Finland. Furthermore, the significance of water services for the economic development of a community cannot be ignored.

**Keywords:** water supply; sanitation; sustainability; resilience; urban development

## 1. Introduction

Water services are a vital part of socio-economic urban development. People need good quality water in their everyday life for drinking, food preparation, and washing. These services are available in many countries, while there are still millions of people living in societies without proper water supply or sanitation. These kinds of unwanted situations are very likely to be more common in the future due to climate change, which will especially impact the most vulnerable people. This is due to enlarging fluctuations in water circumstances from drought to flooding. This causes risks in both the quantity and quality of water.

Proper urban water services require not only an adequate amount of high-quality water resources but also good practices and management in arranging these services, including wastewater services [1–3]. Appropriate institutional frameworks, as well as good governance, are needed to organize adequate water management for a society. When planning urban water services, all aspects of municipal water management should be considered, from abstraction, conveyance, impoundment, storage and processing of raw water (surface water or groundwater), the distribution and supply of purified water for community use, and also the collection and conveyance of community wastewater, treatment of wastewater, and discharging the treated wastewater into surface water. The protection of raw water sources and the environment vis-à-vis the aforesaid activities are also included in water services production. This is a complex system where several external factors have to be taken into account. These factors include meteorological and hydrological conditions, the social and economic situation of the community, technological readiness, and human resources.

Some comprehensive approaches have been developed and applied for sustainable water services. One widely known concept is Integrated Water Resources Management (IWRM), which was introduced at the United Nations Conference on Environment and

Development (UNCED) in Rio de Janeiro in June 1992 for Agenda 21, although many parts of the concept were known and practiced for several decades already. The Global Water Partnership's definition of IWRM states: "IWRM is a process which promotes the coordinated development and management of water, land and related resources, in order to maximize the resultant economic and social welfare in an equitable manner without compromising the sustainability of vital ecosystems" [4].

Closely related to IWRM is the concept of Integrated Urban Water Management, IUWM [5]. According to this approach, sanitation and storm water management should not be planned and implemented separately without acknowledging cross-scale interdependences in freshwater, wastewater, flood control, and storm water. Traditional urban water management has decreased water security and the resilience of urban centers towards, for example, impacts of climate change. IUWM emphasizes the roles of central and local governments, and the views of all stakeholders, including individuals living in the area, are taken into account [5]. Berlin rules [6] have 73 articles concerning the sustainable use of water resources. It also emphasizes the role of individual people, who are the end users of water resources. All stakeholders, from the authorities to the individual water users, should be acknowledged. In Articles 17 (The Right of Access to Water) and 18 (Public Participation and Access to Information), the importance of water services for every individual is justified. This aspect is essential when organizing water services for a community.

In the 1990s in the USA, the water industry sensed that water management was outdated and considered its field from a too narrow perspective. The concept of Total Water Management (TWM) was developed for viewing water management more widely and concerning all types of water use. An idea behind TWM is that the water supply sector is the leader in water resources management. Sustainable development could be promoted by working together to manage water on the basis of natural watersheds. TWM is a concept to create practices for sustainable water management. The focus is not only on water supply and sanitation, it also applies to the whole water sector; supply, wastewater and water quality, agricultural water, hydropower, instream flow management, and security against flood losses. It is closely related to IWRM, and they both emphasize an overall approach to solving water problems [7].

An overall analysis of Finnish water services development in the long term was carried out by Katko [2]. This comprehensive study concentrated on institutional practices during the last decades and compared Finnish practices to those of some other western countries. For example, the Water Poverty Index is introduced by considering five components: water resources; access to resources and services; capacity reflecting socio-economic factors; per capita water use for various purposes; and water quality and environmental impacts [2]. This index can be used when comparing various approaches in practices of urban water services.

The objectives of this study are

- to increase knowledge about organizing sustainable and resilient water services, and
- to gain information on best practices in developing water services.

Two primary research questions have been formulated:

1. What aspects are important in the planning of water services in a community?
2. How should comprehensive water services concerning the whole society be considered in urban development?

This article is based on the doctoral dissertation by Laitinen [3]; some additional conclusions have been made according to the synthesis that was carried out in that research on the sustainability of urban water services. The resilience of urban water services is assessed together with sustainability, and the functionality of water services is viewed using approaches of sustainability and resilience.

In the first part of the article, urban water services are viewed from the perspective of sustainability and resilience. Results are illustrated according to the sequential PESTEL SWOT analysis, which is described in the section on Materials and Methods. Finally, the

results are reviewed and discussed, and some recommendations for Finnish urban water services are given.

## 2. Sustainability, Resilience, and Circular Economy in Urban Water Services

Sustainability and resilience are currently also widely used terms in municipal engineering and urban development. They are closely related, but there can be found some variations in approaches when written about sustainability or resilience. Sustainability has been defined in several previous studies using a combination of three core areas that contribute to the philosophy of sustainable development, economic development, social development, and environmental protection [8]. Hukka and Katko [9] have formulated a definition for the resilience of a water utility based on several definitions of resilience given by various well-known organizations and prominent scholars: "The competence of a utility—also by extending and readjusting its response capabilities and resources needed—to predict, prepare for, adapt to, withstand, communicate and recover promptly, efficiently, and effectively from the consequences of any human-caused intentional and unintentional or naturally occurring hazard, threat or incident—both foreseeable and unexpected—in order to successively produce and maintain safe, reliable and preferred services, to protect the society and the environment, and to learn from the experience gained, now and in the future". Other definitions exist as well, but in this study, the above-mentioned are considered to be applicable concerning urban water services.

Butler et al. [10] identified four types of actions that strengthen a system's sustainability and resilience: mitigation, adaptation, coping, and learning. Koop and Leeuwen [11] analyzed the sustainability of IWRM, and they used five different levels, (i) cities lacking basic water services, (ii) wasteful cities, (iii) water-efficient cities, (iv) resource-efficient and adaptive cities, and (v) water-wise cities. The important aspects of operative action are effective governance, environmental awareness, and community involvement. Inha [12] had a wide view of the resilience of water services in her dissertation. She studied this subject in Finland, India, Nigeria, and the city of Seattle, WA, USA, which gave a very good view of the resilience of water services under different circumstances. She concluded that the resilience factors do not differ much between urban and rural settings, but the difference can be seen according to the development stage of the country or region. Juuti et al. [13] considered local knowledge, a bottom-up approach, good governance, and awareness of historical development as important components when increasing resilience.

The resilience of water services is often connected to the effects of climate change. This is due to increasing seasonal and spatial water scarcity and changing circumstances, which can be resisted by sustainable development and resilient practices in water resources management [3]. Some studies concerning these issues concentrate on the Water-Energy-Food Nexus [14] and have widened it with land use and climate [15].

Aging water infrastructure is nowadays an increasing problem in quite many urban areas (e.g., [1,16,17]). It is one significant issue concerning the resilience of urban water services. Approaches to tackle this are efficient planning and observation [18] and a systematic approach with views on political, financial, technical, and legal control [19]. It is also important to recognize the institutional roles and functions in water services. Katko and Hukka [1] emphasized the distinction between service provision and production, which is a major concern when legislation puts municipalities in charge of providing the services, as the situation is in Finland.

Resource efficiency and circular economy are crucial issues in the policy and development debate on urban development. They are remarkable aspects of planning sustainable urban management, including water management [20]. In urban water management, it is essential to view the whole urban water cycle from water intake, treatment and distribution to wastewater collection, and treatment and discharge back to the waters [21]. One important part of wastewater management, when talking about resource efficiency and circular economy, is the recycling of energy and nutrients, especially phosphorous [22]. Some experiments in wastewater management have also been conducted in source separation of

black and grey water, which could provide better capability for treatment, reuse and heat recovery. This kind of solution has been applied, e.g., in Frankfurt, Germany [23], Qingdao, China [24], and Tampere, Finland [25].

Urban water services form a human-made water cycle within a community. This cycle is related to a natural hydrologic cycle where water comes from the atmosphere down to earth in the form of precipitation, flows to surface waters, infiltrates soil and groundwater, and evaporates, directly or via vegetation, back to the atmosphere. Communities take raw water for domestic and institutional use, purify this water and distribute it via water delivery networks. Wastewater is collected into the sewer network and led to wastewater treatment plants. After the treatment processes, purified wastewater is discharged back to natural waters, where it will return to a part of the hydrologic cycle. This urban water cycle also follows the modes of a circular economy where material and energy are recycled as efficiently as possible. Relationships between these cycles are illustrated in Figure 1 [26].

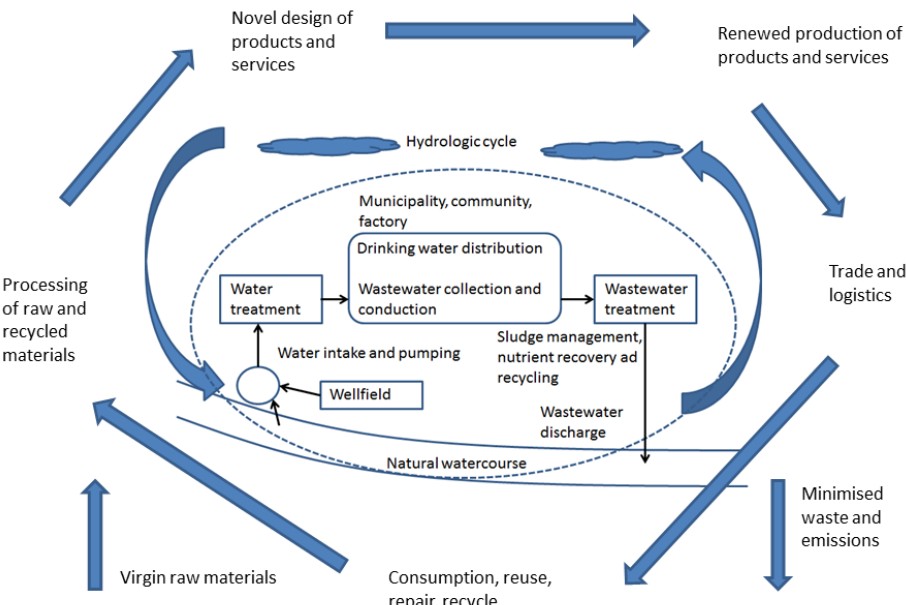

**Figure 1.** Relationships between hydrologic cycle, urban water cycle and principles of circular economy, presented in [26].

## 3. Materials and Methods

The material for this study was collected from previous studies focused on Finnish water services [26–29]. The data were collected using literature reviews, questionnaires, interviews, and different analyses (e.g., life cycle assessment (LCA) and economic viability analysis) for making conclusions from the raw data. To summarize the results, a sequential PESTEL and SWOT analysis was carried out. A PESTEL analysis of Finnish water services has also been carried out earlier [2,30]. In this study, the analysis is extended with SWOT analysis modified from [31,32].

In sequential PESTEL and SWOT analysis, PESTEL is considered a complementary tool to SWOT for looking in detail at external issues. The term PESTEL comes from the domains that are considered in the analysis: Political, Economic, Social, Technological, Environmental, and Legal dimensions. Sequential PESTEL and SWOT analysis are mainly used in specific project strategy and action planning as a basis of decision-making. They have been used in infrastructure and energy projects, especially when evaluating environmental components (e.g., [33–35]. In water initiatives, they have been used, e.g., by Srdjevic et al. [32] in a reconstruction of a water intake structure and Ortega et al. [36] in river basin management.

The PESTEL analysis was used to find out the impact of the following factors [33]:

- Political Factors: including pressures and opportunities brought about by political institutions and the degree of the impact of government policies on the water sector;
- Economic Factors: including economic structures and to what extent the economy impacts decisions can influence the trend for sustainable and resilient water services;
- Social Factors: including cultural components, attitudes, and beliefs that will affect the demand for adequate water services for all;
- Technological Factors: including technological aspects, innovations, barriers, and incentives, and what kind of an impact these have on creating sustainable and resilient water services;
- Legal Factors: laws, regulations, and legislation that will affect the operation of water utilities;
- Environmental Factors: ecological and environmental components that will affect urban water services.

Analytical tools such as PESTEL and SWOT are usually used in strategic planning, decision-making, and action planning [31]. It is noticed that the tool is particularly applicable for identifying the internal and external factors when used sequentially. In urban water, it is significant to assess these factors. Various factors may be difficult to identify as internal or external, or whether their effect is positive or negative. Hence, sequential PESTEL and SWOT analyses were used to recognize all PESTEL factors that affect urban water services. This method is also well suited to finding significant aspects in IUWM. Strengths, weaknesses, opportunities, and threats can be assessed and shown in upcoming challenges in IUWM.

In order to carry out a sequential PESTEL and SWOT analysis, it is recommended to use a group of eight to ten persons [31]. In this study, the data have been analyzed in previous studies by a large group of people, and finally, the sequential PESTEL and SWOT analysis implemented according to those results by the first author of this paper. This can be considered a critical step in this study, but background and various aspects have been properly explored in earlier research that has been carried out on urban water services [26,27,29]. Hence, all opinions and views of those experts have been considered. The idea and a particular scheme of sequential PESTEL and SWOT analysis used in this study and modified from Srdjevic et al. [32] are presented in Figure 2. The process started by studying all PESTEL factors one at a time. When considering the factors, the notes from earlier studies' questionnaires and workshops were reviewed; they were listed and ranked according to their significance. These lists were then used in SWOT analysis to find out strengths, weaknesses, opportunities, and threats in Finnish urban water services. This method was selected because it is simple but an effective way to analyze complex processes in communities.

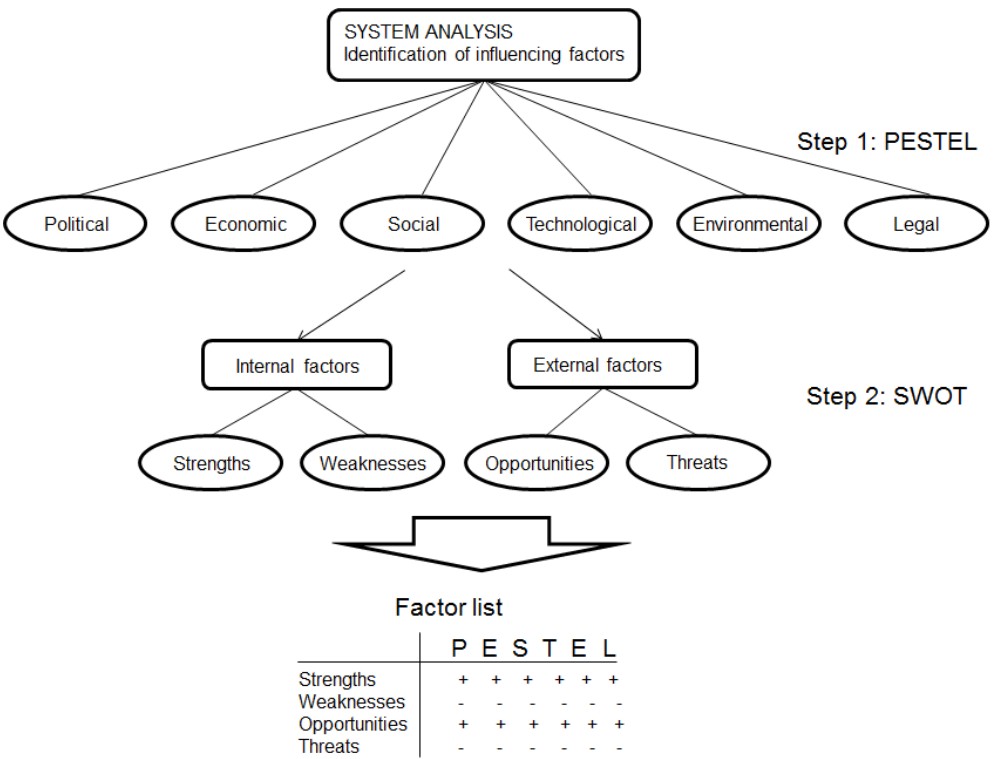

**Figure 2.** A scheme of a sequential PESTEL and SWOT analysis used in this study, modified from Srdjevic et al. [32] by the first author.

## 4. Results

In the conducted PESTEL analysis, factors that are relevant to water services were identified. This process was carried out according to the scope of the PESTEL factors, especially concerning Finnish water services. Conditions in various countries differ so much that it is not possible to do on a global scale. Anyway, the Finnish situation is fairly similar to other western countries.

According to Finnish legislation, municipalities are responsible for providing water services in urban areas. That is the foundation of the institutional configuration of water services. Thus, government and municipal policies are considered a very important political factor in organizing urban water services. Water utilities produce the services [1]; hence, water utilities are relatively independent, and they do their own annual accounting, while their operation and economics are regulated by law. Government resource allocation is important, although it is mainly a matter of the local level due to the municipalities responsible for providing services.

Although water utilities are economically independent, a common economic situation has an impact on water services as well. Water infrastructure is one large part of public assets owned by the municipalities, and for the time being, there is a huge need, especially for the renovation of water and sewer networks, which needs remarkable financial resources. According to the United Nations' SDG 6 Clean Water and Sanitation initiative, proper water services will be provided for everyone by 2030, with special attention to be paid to women, girls, and those living under vulnerable circumstances. While not a major problem in Finland, this is a huge challenge globally.

Migration from rural areas to urban areas is a significant social factor both in land and water use planning. This causes changes in requirements for service provision and both water and sewer networks. On the other hand, multilocality has also become an issue, especially during the COVID-19 pandemic. Water services heavily affect the local environment and population health and well-being, which are very important social components when viewing the effects of sustainable and resilient water services.

New technology is developed all the time that could impact the level of service significantly, or that could be used to achieve future objectives. There is clear development both in water treatment technology and in monitoring and data processing technology. Treatment technology used in water and especially in wastewater treatment plants has been quite the same in Finland during the last 20 years. These plants are designed to remove organic material, suspended solids, and nutrients from wastewater. However, there will also be a demand to purify harmful substances, such as pharmaceuticals and harmful organic matter, in the near future, which cannot be performed properly with currently used technology. New sensor technology, automation, data bases, and modeling will provide new innovations also in IUWM.

Wastewater management is the primary issue in environmental components in water services. The first impact of poor sanitation is locally contaminated water resources, which can dramatically deteriorate the use of the water course. There may also be external issues that will affect wastewater management, such as the impacts of climate change. Urban flooding may cause contamination of a water source, or it may cause sewer overflows, which have severe local impacts on the environment.

For a sustainable society, water services must be equal and equitable for everyone, and this has to be assured by thorough legislation. EU member countries must follow the EU legislation and directives and apply those to their conditions in water management. The legislation will change and become more complicated over the years when member countries apply EU legislation in their own acts.

After identifying the important factors in PESTEL analysis, a SWOT analysis was carried out to brainstorm the external (Threats and Opportunities) and internal (Weaknesses and Strengths) categories considering and reflecting those on the objectives and outcomes of the study. In this stage, the results of PESTEL factors are the starting point. The factors were ranked (scale: very important, important, not important, and ignored) considering their potential impact of them on the objectives and outcomes and the likelihood of such impacts. The results of the analysis are first listed, and after that, as a conclusion, in Table 1 all the factors are illustrated in a SWOT table. There are some factors that may be considered threats or opportunities or weaknesses or strengths at the same time, depending on the local circumstances.

**Table 1.** SWOT table of 'Sustainable and resilient urban water services from the point of view of water utility".

| | Favorable for Achieving the Objectives | Unfavorable for Achieving the Objectives |
|---|---|---|
| | **Opportunities** | **Threats** |
| **External** | • Governmental and municipal policies<br>• Stakeholder needs<br>• Economic situation<br>• Management and operation<br>• Educated and skilled personnel<br>• Future legislation, international agreements and cooperation<br>• New technologies<br>• Health and environmental issues | • Municipal resource allocations<br>• Lobbying<br>• Climate change<br>• Population structure<br>• Existing water infrastructure<br>• Technological innovations |
| | **Strengths** | **Weaknesses** |
| **Internal** | • Political stability<br>• Economic and financial system and resources<br>• Institutional framework and good governance<br>• Educated and skilled personnel<br>• Awareness<br>• Water infrastructure<br>• Data management and smart water systems | • Deteriorating water pipe and sewer networks<br>• Treatment of harmful substances<br>• A large number of small water utilities |

When comparing the results with some other studies (e.g., [10,37,38]) on sustainable and resilient water services using other methods, it can be seen that they come to the same kind of conclusions. This indicates that chosen sequential PESTEL and SWOT method suits reasonably well when assessing how advanced urban water services should operate in resilient societies. In the above-mentioned studies, risks, uncertainty, and reliability are emphasized; in this study, we have concentrated more on practices implemented in water utilities and in the institutional setup.

Governmental and municipal policies, as well as economic situations, are considered opportunities, though they may also be threats. In Finland, awareness of the importance of adequate water services in all stakeholder groups is very high, and this can be considered a prerequisite to good governance and policy. The economic situation is also considered an opportunity because the principle of full cost recovery is accepted as an important component of strengthening the economy of all water utilities in the country. In order to keep health and environmental issues as opportunities, proper understanding and cooperation between several sectors are needed: land use, industry, agriculture, tourism, recreation, and transportation.

The primary threats to Finnish water services include deterioration of water infrastructure, climate change, and municipal resource allocation. Although water supply and sanitation should be an independent sector within municipal services, it is sometimes a part of municipal administration without autonomy. Pricing policy should be completely defined according to the needs of water services, including all investments as well as operation and management costs. This sounds simple, but still, water infrastructure, especially water supply and sewer networks, has deteriorated to an unacceptable level in several urban areas. Pricing of water and water infrastructure investments has not necessarily been planned according to the needs of water services but according to political preferences. The Finnish Water Services Act allows the owners of water utilities to make a reasonable profit. This profit is not defined, and in some cases, it can be considered excessive.

The institutional framework is a very important factor in organizing sustainable and resilient water services for a community. Public-private cooperation works well in a flexible combination of public responsibility and strong private sector know-how. This needs political stability, which can be considered to be the situation in Finland as well as in several other western countries. In a previous study [29], stakeholders assessed that educated and skilled personnel is a very important factor for an adequately operational water utility, which is considered a strength in the Finnish water sector in this study.

Deteriorating water pipe and sewer networks and treatment of harmful substances are the main weaknesses in Finnish urban water management at the moment. Furthermore, the large amount of water utilities is a weakness because of the large amount of very small utilities without adequate expertise or skilled personnel. However, it is more a question of how well the utility is managed and operated rather than the size itself. The results of this study produced a new knowledge base on how to develop the governance and practices of urban water services. The main problems that have been raised cannot be solved only by water utilities. The aspects given in this study must also be adopted by authorities, municipalities, and political decision makers.

## 5. Discussion

The primary focus of this study was finding key aspects in urban water services for providing healthy, environmentally secure services at a reasonable cost. The crucial concepts as the basis of the approach were IWRM and IUWM. Sustainability and resilience in urban water services were the key issues in this study. The results were analyzed and disseminated with a sequential PESTEL and SWOT analysis. Key conclusions can be summarized in (a) proper data, knowledge and asset management, (b) resource efficiency and circular economy issues, (c) principles of green economy, and (d) good governance and competent staff.

The outcome that was targeted in the objectives of this study was "Sustainable and resilient urban water services". Pillars that support and have an effect on this outcome are illustrated in Figure 3.

**Figure 3.** Pillars of sustainable and resilient water services [3].

The research questions set at the beginning of the study are about the important factors in planning urban water services and the role of sustainable and resilient water services in urban development.

A fundamental issue in urban water services is to ensure safe water continuously and at a reasonable price so that the operation causes no harm to the environment. The chief components to implement this in a sustainable way and provide resilient service include good governance and institutional setup, economic sustainability, asset management, and safeguarding of competence. Although large water utilities typically have the best possibility to maintain good expertise, the economies of scale cannot directly be applied to water services. There are several examples of properly operating water utilities that are small- or medium-sized enterprises.

Water services are an important issue in urban development. It has to be taken into account in land use planning and municipal engineering, and vice versa. Water services of a community should be taken into account at all levels of decision-making, due to its nature as an essential commodity in society.

## 6. Conclusions

The recommendation made in [3] includes institutional development; good governance; viable pricing policy; educated and competent staff; technological development; and proper data and knowledge management. The actions are quite valid in the long run, and they are also possible to implement at a reasonable cost. Water services are a critical part of a society, and it has to be resilient also in uncertain times. This can be seen, for example, during the COVID-19 pandemic and wartimes. Of course, sustainability and resilience of water services have to be organized considering a normally operating society, but well-planned and operating services also work well in uncertain times. For the resilient operation of water services, it is very important to have viable technology and equipment, as well as a competent maintenance system for keeping the complex system working continuously. However, this also requires an advanced institutional framework and systematic capacity building throughout the water sector. Water supply and sanitation

are part of comprehensive water services, but although it is considered to be the most important type of water use in society, other forms of water use must be taken into account when organizing municipal water services. Climate change and the changing community structure pose new challenges to urban water services, and managing their impact should be the next step in the development of the sector.

The sustainability and resilience of water services have been studied, but it still needs more systematic research by universities and other research institutes. This subject is interdisciplinary and dependent on several other factors in urban and social planning. There are various possible approaches that can be chosen and followed to develop this essential service for human settlements in the future.

**Author Contributions:** Conceptualization, J.L.; methodology, J.L., T.S.K. and J.J.H.; investigation, J.L.; writing—original draft preparation, J.L..; writing—review and editing, T.S.K., J.J.H., P.J. and R.J.; supervision, T.S.K., P.J. and J.J.H.; project administration, J.L. All authors have read and agreed to the published version of the manuscript.

**Funding:** This work was supported by Maa- ja vesitekniikan tuki ry (Finnish support foundation).

**Institutional Review Board Statement:** Not applicable.

**Informed Consent Statement:** Not applicable.

**Data Availability Statement:** The data used in this study is available in data sets of Finnish Environment Institute.

**Acknowledgments:** The authors thank the peer reviewers and the editors for valuable comments and feedback.

**Conflicts of Interest:** The authors declare no conflict of interest.

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
