# Peer review of "Governance and Practices for Achieving Sustainable and Resilient Urban Water Services"

_water, doi:10.3390/w14132009_

Round 1

Reviewer 1 Report

This article includes errors of a stunning amateurism, especially considering that it targets a highly indexed journal:

  • Please see the rows from 380 to 390. It is copy/paste from the Instructions for authors

References

References must be numbered in order of appearance in the text (including citations in tables and legends) and listed individually at the end of the manuscript. We recommend preparing the references with a bibliography software package, such as EndNote, Reference Manager or Zotero to avoid typing mistakes and duplicated references. Include the digital object identifier (DOI) for all references where available.

Citations and references in the Supplementary Materials are permitted provided that they also appear in the reference list here.

In the text, reference numbers should be placed in square brackets [ ] and placed before the punctuation; for example [1], [1–3] or [1,3]. For embedded citations in the text with pagination, use both parentheses and brackets to indicate the reference number and page numbers; for example [5] (p. 10), or [6] (pp. 101–105).

It's incredibly horror!!!

The rest of the article falls into the same level of very low quality.

  • The Conclusions section are very general and without scientific consistency, beyond the fact that they have only a few lines.
  • The sources must be added under each table and figure.
  • Figure 1. Relationships between hydrologic cycle, urban water cycle and principles of circular economy, presented in [26] is full copy from other authors? Do you have the Copyrights in this regards?

- Authors should take into consideration much more recent publications in the sphere of discussed subject matter, especially studies conducted during the last 5 years. There is no Literature review section, only few references and without a relevant coherence of the quotation in the text.

- Human proofreading, English grammar and spelling correction are also required in order to improve the quality of the manuscript.

 These are just some of the reasons why this article deserves rejection without the possibility of resubmission.

Author Response

Dear reviwer,

Thank you for commenting our manuscript for MDPI Water. It is obvious that you did not like it, but due to other comments we are revising and resubmitting the article. We would like to give some response to your comments.

Figure 1. Relationships between hydrologic cycle, urban water cycle and principles of circular economy, presented in [26] is full copy from other authors? Do you have the Copyrights in this regards?

Response: Figure 1 is formed by the first author and published in one of his earlier articles (there is also reference number given), so there is no problem with Copywright.

Human proofreading, English grammar and spelling correction are also required in order to improve the quality of the manuscript.

Response: English grammas and spelling correction have been done by a native English speaker and a language professional.

Please see the rows from 380 to 390. It is copy/paste from the Instructions for authors.

Response: The mistake in References chapter is not a copy/paste, we just forgot to delete that part from the given Word template given by MDPI. We used it as instructed, but unfortunately that oblivion was a human error.

You may also want to see the comments of the other reviewers, we add those here:

Reviewer 2:

Abstract:

overall good, but: " In our study, we used a sequential PESTEL SWOT analysis for reviewing results of previous studies.

Introduction:

Line 32: "Appropriate institutional frameworks, as well as good governance, are needed to organize adequate water management for a society." 

Line 90: "This study" ... which? this article? or another one you are reviewing in the article - unclear, so please rephrase

Lines 96/97: "Results are illustrated according to the sequential PESTEL SWOT analysis, which is described in the section on Material and Methods."

Materials and Methods:

Lines 115-116: "Other definitions exist as well, but in this study REMOVE COMMA HERE  the above mentioned are considered to be applicable concerning urban water services."

Line 154: "Urban water services form a human-made water cycle within a community."

Line 170: "The material for this study was collected from previous studies focused on Finnish water services."

Line 200: "Analytical tools such as PESTEL and SWOT are usually used in strategic planning, decision-making and action planning."

Line 215:  "The idea and a particular scheme of sequential PESTEL and SWOT analysis used in this study and modified from Srdjevic et al. [32] REMOVE THE COMMA HERE are presented in Figure 2. "

Results: 

Line 222: "In the conducted PESTEL analysis, factors that are relevant in water services were identified."

Discussion:

ok

Conclusions:

ok

References:

REMOVE THIS: References must be numbered in order of appearance in the text (including citations in tables and legends) and listed individually at the end of the manuscript. We recommend preparing the references with a bibliography software package, such as Water 2022, 14, x FOR PEER REVIEW 11 of 12
EndNote, ReferenceManager or Zotero to avoid typing mistakes and duplicated references. Include the digital object identifier 
(DOI) for all references where available. 
Citations and references in the Supplementary Materials are permitted provided that they also appear in the reference list here. 

In the text, reference numbers should be placed in square brackets [ ] and placed before the punctuation; for example [1], [1–3]
or [1,3]. For embedded citations in the text with pagination, use both parentheses and brackets to indicate the reference number
and page numbers; for example [5] (p. 10), or [6] (pp. 101–105).

[1] Katko, T.S. and Hukka, J.J., 2015. Social and Economic Importance of Water Services for the Built Environment: Need for More 3Structured Thinking. 8th Nordic Conference on Construction Economics and Organization. Procedia Economics and Finance
21:217. https://doi.org/10.1016/S2212-5671(15)00170-7.

[10] Butler, D., Ward, S., Sweetapple, C., Astaraie-Imani, M., Diao, K., Farmani, R. and Fu, G., 2016. Reliable, resilient and sustainable water management: the Safe & SuRe approach. Global Challenges, Vol. 1(1). REMOVE: Research article: Water 2016, pp. 63-77

[21] Arden, S., Ma, X. and Brown, M., 2018. Holistic analysis of urban water systems in the Greater Cincinnati region: (2) resource 
use profiles by emergy accounting approach. Water Research  X SPECIFY!, Vol. 2, Art. Nr 100012. pp. 1-9.

[25] Malila, R., Lehtoranta, S. and Viskari, E-L., 2019. The role of source separation in nutrient recovery – comparison of alternative
wastewater systems. Journal of Cleaner Production 219(1), REMOVE pp. 350-358.

[28] Laitinen, J., Moliis, K. and Surakka, M.. 2017. Resource efficient wastewater treatment in a developing area – Climate change 
impacts and economic feasibility. Ecological Engineering 103, 217-225. 

[29] Laitinen, J., Kallio, J., Katko, T.S., Juuti, P. and Hukka, J.J.. 2020. Resilient water services for the 21st century society – stakeholder survey in Finland. Water REMOVE, 2020, 12, 187, 1-12

[37]  WHY IS THIS HERE - OR NOT COMPLETED?

Reviewer 3:

Dear authors

Many thanks for putting together such an exciting piece of research. The study reports interesting findings that will be useful for the research community; however, some issues must be addressed prior to the full acceptance of this paper.

There are many formatting, font types, font size issues across the paper, and some inconsistencies in the references. In addition, the abstract is structured oddly and does not flow well. A good trick is to plan your argument in 6 sentences and then use these to structure your abstract:

  1. Introduction. Describe what topic your paper covers. Provide the reader with a background to the study. Avoid unnecessary content.
  2. State the problem. What is the key research question? Again, in one sentence.
  3. Summarise why nobody else has adequately answered the research question yet. Emphasise the gap in the literature. You could use a phrase such as “Previous work has failed to address...”.
  4. Explain how you have approached the research question. What’s your big new idea?
  5. In one sentence, describe how are you planning to go about doing the research? Provide an outline of the methods you used. Did you run experiments? Carry out case studies? Design experiments? Interviews?
  6. What is the key impact of your research? What conclusions did you draw or are you expecting to draw and what are the implications? What is the primary take-home message?

The introduction provides a good overview of the problem under investigation. I believe the paragraph in L52 and paragraph in L60 can be merged. It would be good to obtain more information on the Berlin rules and why they are of relevance to this study. The paragraphs below L76 would need some re-work. I find this part a little hectic and hard to follow. There isn’t a systematic flow of ideas.

The materials and methods described are well, but we’ll need most of the literature presented to be moved to a dedicated section (L100-165). I think we’ll need to separate what is the theory from what is the method adopted. In addition, we will need to know much more in terms of the processes used and adopted. Within PESTEL and SWOT, what are the critical steps adopted in this study?

The results section is well-structured and clear. As for the discussion, the key points raised are pertinent and intriguing; however, how have these results contributed to the study? How has this information enriched the planning and water management systems in place? What do we need to change for them to become better?

The conclusion is short, and we’ll need a more robust finale. For instance, what sort of limitations were found in this study and what are the next steps?

We hope these changes will make our manuscript better.

The authors

Reviewer 2 Report

Abstract:

overall good, but: " In our study, we used a sequential PESTEL SWOT analysis for reviewing results of previous studies.

Introduction:

Line 32: "Appropriate institutional frameworks, as well as good governance, are needed to organize adequate water management for a society." 

Line 90: "This study" ... - which? this article? or another one you are reviewing in the article - unclear, so please rephrase

Lines 96/97: "Results are illustrated according to the sequential PESTEL SWOT analysis, which is described in the section on Material and Methods."

Materials and Methods:

Lines 115-116: "Other definitions exist as well, but in this study REMOVE COMMA HERE  the above mentioned are considered to be applicable concerning urban water services."

Line 154: "Urban water services form a human-made water cycle within a community."

Line 170: "The material for this study was collected from previous studies focused on Finnish water services."

Line 200: "Analytical tools such as PESTEL and SWOT are usually used in strategic planning, decision-making and action planning."

Line 215:  "The idea and a particular scheme of sequential PESTEL and SWOT analysis used in this study and modified from Srdjevic et al. [32] REMOVE THE COMMA HERE are presented in Figure 2. "

Results: 

Line 222: "In the conducted PESTEL analysis, factors that are relevant in water services were identified."

Discussion:

ok

Conclusions:

ok

References:

REMOVE THIS: References must be numbered in order of appearance in the text (including citations in tables and legends) and listed individually at the end of the manuscript. We recommend preparing the references with a bibliography software package, such as Water 2022, 14, x FOR PEER REVIEW 11 of 12
EndNote, ReferenceManager or Zotero to avoid typing mistakes and duplicated references. Include the digital object identifier 
(DOI) for all references where available. 
Citations and references in the Supplementary Materials are permitted provided that they also appear in the reference list here. 

In the text, reference numbers should be placed in square brackets [ ] and placed before the punctuation; for example [1], [1–3]
or [1,3]. For embedded citations in the text with pagination, use both parentheses and brackets to indicate the reference number
and page numbers; for example [5] (p. 10), or [6] (pp. 101–105).

[1] Katko, T.S. and Hukka, J.J., 2015. Social and Economic Importance of Water Services for the Built Environment: Need for More 3Structured Thinking. 8th Nordic Conference on Construction Economics and Organization. Procedia Economics and Finance
21:217. https://doi.org/10.1016/S2212-5671(15)00170-7.

[10] Butler, D., Ward, S., Sweetapple, C., Astaraie-Imani, M., Diao, K., Farmani, R. and Fu, G., 2016. Reliable, resilient and sustainable water management: the Safe & SuRe approach. Global Challenges, Vol. 1(1). REMOVE: Research article: Water 2016, pp. 63-77

[21] Arden, S., Ma, X. and Brown, M., 2018. Holistic analysis of urban water systems in the Greater Cincinnati region: (2) resource 
use profiles by emergy accounting approach. Water Research  X SPECIFY!, Vol. 2, Art. Nr 100012. pp. 1-9.

[25] Malila, R., Lehtoranta, S. and Viskari, E-L., 2019. The role of source separation in nutrient recovery – comparison of alternative
wastewater systems. Journal of Cleaner Production 219(1), REMOVE pp. 350-358.

[28] Laitinen, J., Moliis, K. and Surakka, M.. 2017. Resource efficient wastewater treatment in a developing area – Climate change 
impacts and economic feasibility. Ecological Engineering 103, 217-225. 

[29] Laitinen, J., Kallio, J., Katko, T.S., Juuti, P. and Hukka, J.J.. 2020. Resilient water services for the 21st century society – stakeholder survey in Finland. Water REMOVE, 2020, 12, 187, 1-12

[37]  WHY IS THIS HERE - OR NOT COMPLETED?

you may want to have a look at the attached article also

Author Response

Dear reviewer,

Thank you very much for reviewing our manuscript and your encouraging comments. Please find responses to your comments:

There are several spelling mistakes that you have dound.

Response: Thank you for your attention reading, we have now corrected those mistakes.

References

Response: The mistake in References chapter is due to forgetting to delete that part from the given MDPI Word platform. We have now deleted it. We made also corrections that you have found in our references list.

We hopethese responses and changes will be such that you hoped for making this article better.

The authors

Reviewer 3 Report

Dear authors

Many thanks for putting together such an exciting piece of research. The study reports interesting findings that will be useful for the research community; however, some issues must be addressed prior to the full acceptance of this paper.

There are many formatting, font types, font size issues across the paper, and some inconsistencies in the references. In addition, the abstract is structured oddly and does not flow well. A good trick is to plan your argument in 6 sentences and then use these to structure your abstract:

  1. Introduction. Describe what topic your paper covers. Provide the reader with a background to the study. Avoid unnecessary content.
  2. State the problem. What is the key research question? Again, in one sentence.
  3. Summarise why nobody else has adequately answered the research question yet. Emphasise the gap in the literature. You could use a phrase such as “Previous work has failed to address...”.
  4. Explain how you have approached the research question. What’s your big new idea?
  5. In one sentence, describe how are you planning to go about doing the research? Provide an outline of the methods you used. Did you run experiments? Carry out case studies? Design experiments? Interviews?
  6. What is the key impact of your research? What conclusions did you draw or are you expecting to draw and what are the implications? What is the primary take-home message?

The introduction provides a good overview of the problem under investigation. I believe the paragraph in L52 and paragraph in L60 can be merged. It would be good to obtain more information on the Berlin rules and why they are of relevance to this study. The paragraphs below L76 would need some re-work. I find this part a little hectic and hard to follow. There isn’t a systematic flow of ideas.

The materials and methods described are well, but we’ll need most of the literature presented to be moved to a dedicated section (L100-165). I think we’ll need to separate what is the theory from what is the method adopted. In addition, we will need to know much more in terms of the processes used and adopted. Within PESTEL and SWOT, what are the critical steps adopted in this study?

The results section is well-structured and clear. As for the discussion, the key points raised are pertinent and intriguing; however, how have these results contributed to the study? How has this information enriched the planning and water management systems in place? What do we need to change for them to become better?

The conclusion is short, and we’ll need a more robust finale. For instance, what sort of limitations were found in this study and what are the next steps?

Author Response

Dear reviewer,

Thank you very much for reviewing our manuscript and giving valuable ideas how to make it better. We have done editing according to your comments, here are our responses:

...the abstract is structured oddly and does not flow well. A good trick is to plan your argument in 6 sentences and then use these to structure your abstract:

Response: Thank you for a structured trick, we changed the Abstract according to your advice.

The introduction provides a good overview of the problem under investigation. I believe the paragraph in L52 and paragraph in L60 can be merged. It would be good to obtain more information on the Berlin rules and why they are of relevance to this study. The paragraphs below L76 would need some re-work. I find this part a little hectic and hard to follow. There isn’t a systematic flow of ideas.

Response: We merged those two paragraph, it is better that way. We explained shortly the role of Berlin rules more, we did not want to emphasize it too much, but as a basis of water services we wanted to mention it.

The materials and methods described are well, but we’ll need most of the literature presented to be moved to a dedicated section (L100-165). I think we’ll need to separate what is the theory from what is the method adopted. In addition, we will need to know much more in terms of the processes used and adopted. Within PESTEL and SWOT, what are the critical steps adopted in this study?

Response: We thought that it is a good idea to separate a new section of literature and theory of sustainability, resilience and circular economy between Introduction and Material and methods. In Material and methods chapter we exlpained more about use of sequential PESTEL and SWOT analysis.

The results section is well-structured and clear. As for the discussion, the key points raised are pertinent and intriguing; however, how have these results contributed to the study? How has this information enriched the planning and water management systems in place? What do we need to change for them to become better?

Response: We addad some reflection of the results so that it is easier to find out how this information enrich the planning and water management.

The conclusion is short, and we’ll need a more robust finale. For instance, what sort of limitations were found in this study and what are the next steps?

Response: We added more detailed conclusions and some reflection how urban water services need more attention not only in technological development, but also in institutional development and capacity building. Also security, governance and good practices have to be observed.

We hopethese responses and changes will be such that you hoped for making this article better.

The authors

Round 2

Reviewer 1 Report

The attitude of the authors is incredible in the context of academic research. It is inconceivable and shows a total lack of ethics that the authors justify themselves by including the comments of other reviewers.

Authors send me reply to my own comments such as: You may also want to see the comments of the other reviewers, we add those here ….”. So the authors try to influence my opinion by coercion and self-praise, in order to obtain easy acceptance without any considerable improvement of the scientific quality of this article???

Each reviewer is independent and the authors should not try to influence their decisions.

Second, the authors explaination „The mistake in References chapter is not a copy/paste, we just forgot to delete that part from the given Word template given by MDPI. We used it as instructed, but unfortunately that oblivion was a human error. denotes a shocking amateurism and self-sufficiency. There is no excuse for a superficial research article.

MDPI journals have a great reputation, so it is not normal to try to tarnish this scientific credibility with very poor quality and disastrously written articles, because we are not talking about high school newspapers or magazines.

On the other hand, the authors ignored most of the recommendations in the previous review. For instance, not a single new reference was included în order to improve quality, still no Literature review section etc.

I recommend rejection once again without the possibility of resubmission.

Author Response

Dear referee,

I am sorry our manuscript does not please you. We cannot do all changes that would please you in this ready written paper, but we added some references for comparing to other studies concerning also the same subject. We still do not have a special literature review section, our literature review is in sections 1 and 2.

We hope these will give some approvements to the manuscript in your opinion.

Best regards, the authors

Reviewer 3 Report

Dear authors

Many thanks for addressing the comments. The manuscript reads and flows much better. The content added also enabled us to understand better some of the theoretical and methodological approaches as well as the results of this project.

Author Response

Dear referee,

Thank you very much for your nice and supportive comments. We have still done some changes by adding 3 more references for comparing our results to the results of these previous studies on the same subject.

Best regards, the authors